# Evaluating Predictive Models of Tree Foliar Moisture Content for Application to Multispectral UAS Data: A Laboratory Study

Lauren E. Lad [1,*], Wade T. Tinkham [2], Aaron M. Sparks [3] and Alistair M. S. Smith [4]

1 Department of Forest and Rangeland Stewardship, Colorado State University, 1472 Campus Delivery, Fort Collins, CO 80523, USA

2 Rocky Mountain Research Station, United States Department of Agriculture Forest Service, 240 W Prospect Road, Fort Collins, CO 80526, USA; wade.tinkham@usda.gov

3 Department of Forest, Rangeland, and Fire Sciences, University of Idaho, 875 Perimeter Drive MS 1133, Moscow, ID 83844, USA; asparks@uidaho.edu

4 Department of Earth and Spatial Sciences, University of Idaho, 875 Perimeter Drive MS 3025, Moscow, ID 83844, USA; alistair@uidaho.edu

* Correspondence: lauren.lad@colostate.edu

**Abstract:** Water supply is a critical component of tree physiological health, influencing a tree's photosynthetic activity and resilience to disturbances. The climatic regions of the western United States are particularly at risk from increasing drought, fire, and pest interactions. Existing methods for quantifying drought stress and a tree's relative resilience against disturbances mostly use moderate-scale (20–30 m) multispectral satellite sensor data. However, tree water status (i.e., water stress) quantification using sensors like Landsat and Sentinel are error-prone given that the spectral reflectance of pixels are a mixture of the dominant tree canopy, surface vegetation, and soil. Uncrewed aerial systems (UAS) equipped with multispectral sensors could potentially provide individual tree water status. In this study, we assess whether the simulated band equivalent reflectance (BER) of a common UAS optical multispectral sensor can accurately quantify the foliar moisture content and water stress status of individual trees. To achieve this, water was withheld from groups of Douglas-fir and western white pine saplings. Then, measurements of each sapling's foliar moisture content (FMC) and spectral reflectance were converted to BER of a consumer-grade multispectral camera commonly used on UAS. These bands were used in two classification models and three regression models to develop a best-performing FMC model for predicting either the water status (i.e., drought-stressed or healthy) or the foliar moisture content of each sapling, respectively. Our top-performing models were a logistic regression classification and a multiple linear regression which achieved a classification accuracy of 96.55% and an r$^2$ of 82.62, respectively. These FMC models could provide an important tool for investigating tree crown level water stress, as well as drought interactions with other disturbances, and provide land managers with a vital indicator of tree resilience.

**Keywords:** drought; conifer foliar moisture; drone; UAV

## 1. Introduction

Foliar moisture content (FMC) is a key indicator of vegetation health and influences individual plant resilience to weather, climatic variability, and disturbances such as insects, disease, and fire [1,2]. Further, foliar moisture content changes seasonally and daily based on evapotranspiration and water loss, or through precipitation and water uptake [1–3]. Beyond indicating the relative ratio of dry and wet plant material, foliar moisture content informs estimates of fire risk and rate of spread [4]. Additionally, foliar moisture plays a key role in the feedback loops of disturbances, such as bark beetles, by influencing the likelihood of infestation and resources available for defense following infestation [5]. Water stress on trees in California, US, recently resulted in >100 million trees succumbing to beetle infestation due to reduced defense mechanisms related to foliar moisture [6]. Lower limit

thresholds of foliar moisture also indicate when vegetation is vulnerable to disturbance-induced damage and mortality [7]. Lower foliar moisture content values are related to decreased resilience and increased mortality from both fire and insects [8,9]. Conventional measures of FMC require the collection of foliage from each plant and a comparison of the foliage's dry and wet weight [4,10], providing a sample of the data to represent an entire population. The need to oven-dry specimens can delay the applicability of foliar moisture observations after collection, which limits its use for rapid hazard assessment applications [7]. Although in other parts of the fuel stratum (i.e., woody debris, duff, and litter), there has been the development of rapid sensing systems [11] and research to understand the drivers of moisture content [12,13], limited research has focused on foliar moisture content [3]. As such, there is limited capacity to collect rapid, spatially continuous measures of FMC, particularly at scales and extents relevant to management decision-making (i.e., 10 s of hectares).

One alternative method of assessing FMC continuously and across larger areas is the use of aircraft and satellite-based remote sensing [14]. The Normalized Difference Moisture Index (NDMI), which uses near-infrared (NIR) and short-wave infrared (SWIR) spectral bands, is used to monitor drought because of the index's sensitivity to changes in plant water content [15]. Originally, this index was named the Normalized Difference Water Index (NDWI) but is generally referred to by NDMI following the development of a different NDWI index by McFeeters (1996) [16]. The NDMI was developed using laboratory measurements of reflectance and has since demonstrated its utility in characterizing vegetation drought stress in diverse ecosystems including forested areas [15] and peatlands [17]. At multiple scales and across several conifer species, NDMI and red edge spectral wavelengths have been demonstrated to have a significant correlation with plant water content [18,19]. Seong et al. (2015) used NDMI to assess drought stress in forests in Korea and found that areas with increased NDMI values, indicative of drought-stress, experienced large forest fires, thus showing a causal effect between low FMC and higher fire risk [20]. From satellite observations, NDMI has been demonstrated as a better predictor of live FMC than the more commonly applied Normalized Difference Vegetation Index (NDVI) [21], suggesting its potential for improving coarse-scale management of drought in forests. However, application of NDMI through satellite-based remote sensing is limited due to its coarse spatial and temporal resolutions. Landsat and Sentinel-2 satellites collect imagery at 30 m and 20 m spatial resolution, respectively, and at 16- and 5-day return intervals. Quantifying FMC using moderate spatial resolution imagery can be problematic because spectral reflectance from individual pixels is a mixture of canopy and understory components, which makes FMC estimation at the plant level challenging. Further, the low (relative to UAS) temporal resolution of these satellites limits the timely assessment of FMC, which varies diurnally [1]. These limitations in resolution restrict its application to inform management decisions that require rapid, timely information such as responding to wildfire incidents or informing early warning systems [7].

At the other end of the spatial resolution spectrum, fine-scale laboratory and field experiments can provide insight into stress impacts on individual plant physiology, growth, and mortality. Several studies have used toxicological dose-response experiments to assess how varying levels of stressors, such as drought and fire, affect conifer sapling physiology and whether these changes can be detected using remote sensing data. Sparks et al. (2016) evaluated the impact of increasing fire radiative energy (FRE), or the total radiative heat flux from surface fires, on *Pinus contorta* var. *latifolia* and *Larix occidentalis* saplings and whether physiological responses could be accurately quantified using foliar spectral reflectance [22]. They found that the change in NDVI (dNDVI) from pre- to post-fire could accurately quantify metrics of physiological stress including reduced net photosynthesis and chlorophyll fluorescence [22]. Other spectral indices such as the Photochemical Reflectance Index (PRI) [23] have also been shown to accurately quantify tree stress. Specifically, Sparks et al. [24] observed that PRI accurately quantified reductions in net photosynthesis and chlorophyll fluorescence in *Pinus monticola* and *Pseudotsuga menziesii* saplings sub-

jected to fires of varying intensity [24]. Additionally, Partelli-Feltrin et al. [8] examined the interaction of drought-stressed ponderosa pine (*Pinus ponderosa* Lawson and C. Lawson) saplings and FRE doses in a laboratory, finding that drought-stressed saplings died at lower FRE doses than well-watered saplings. These studies demonstrate the utility of fine-scale laboratory experiments for examining linkages between spectral indices derived from foliar spectral reflectance (i.e., NDVI, PRI) and plant physiological metrics (i.e., net photosynthesis, chlorophyll fluorescence, FMC) [8]. Based on previous laboratory studies, there is a clear linkage between pre-fire conifer sapling water stress and a sapling's resilience to fire induced tree mortality [8,22]. However, few tree species and size classes have been assessed, restricting the application of these findings to inform management decision-making processes.

Uncrewed aerial systems (UAS) offer a potential bridge between fine-scale laboratory and field sampling efforts and satellite-derived landscape-scale assessments by providing fine-spatial resolution continuous data at the forest-stand scale [25]. Prior to 2023, UAS were commonly described as unmanned aerial systems, but a change in terminology by the United States Pentagon and the United States National Oceanic and Atmospheric Administration (NOAA) in late 2022 led to the widely adopted shift from unmanned to uncrewed. Specifically, UAS have demonstrated their ability to characterize individual tree attributes, including stem diameter and height, in fire-adapted moderate canopy closure *Pinus ponderosa* forests and provide users with control of the temporal resolution of imagery [26]. A growing range of UAS sensors, such as the MicaSense Dual-Camera (MicaSense, Seattle, WA, USA), provide spectral information not available on Landsat and Sentinel, including three red edge bands [27,28]. These narrower red edge bands can facilitate early detection of physiological stress and improve the scale of stress-detection through increased spatial resolution compared to satellite imagery [29]. Using high-resolution satellite imagery from RapidEye, Eitel et al. [19] showed that a spectral index using wavelengths in the red edge was able to accurately identify stressed trees in a pinon–juniper woodland 16 days earlier than NDVI, highlighting the utility of red edge spectra for early stress detection. NDVI, and other indices derived from NIR, red, green, and blue spectral bands, have strong potential to predict stress when used as training data in random forest and support vector machine classification models [30]. When comparing a UAS-derived NDVI and National Agricultural Imagery Program imagery for classifying trees into five health classes, the UAS data out-predicted NAIP by 14.97% [30]. However, while indices such as NDVI have proven utility in detecting forest stress, these indices may not be sensitive to the rapid changes in tree moisture that result from environmental stressors [31]. Thus, this study examines a variety of remotely sensed indices to identify which indices may produce robust predictions of drought stress and status.

In this study, the overall objective was to assess if band equivalent reflectance (BER) of a consumer-grade multispectral UAS camera could be used to predict FMC and sapling drought-stress, defined here as saplings with an FMC lower than 120%, for two western United States conifer species: western white pine (*Pinus monticola* Douglas ex D. Don) and Douglas-fir (*Pseudotsuga menziessii* (Mirb.) Franco var. *glauca* (Beissn.) Franco). To address this objective, directly measured FMC of saplings (*n* = 123) at varying levels of drought stress was collected concurrently with sapling foliar spectral reflectance acquired with a spectroradiometer. This spectral reflectance data were used as input data for classification and regression models to predict FMC and drought stress status.

## 2. Materials and Methods

### 2.1. Saplings and Study Treatments

Saplings were grown in a climate-controlled greenhouse at the University of Idaho, Moscow, Idaho. Detailed information on sapling growth and storage is described by Smith et al. [32]. A total of 62 western white pine and 61 Douglas-fir saplings were grown for 2 years in 9.5-L pots in the greenhouse before being relocated to the Idaho Fire Initiative for Research and Education (IFIRE) combustion laboratory. Once at the IFIRE lab, both

species were randomly divided into a control and three drought groups to provide a range of FMC (Table 1). Before the experiment, all seedlings were watered to field capacity daily; then, beginning 25 days before spectral reflectance and FMC measurements, the three drought groups had water withheld for progressively shorter intervals, while the control was watered to field capacity daily. Specifically, water was withheld from Group 3 for 25 days, 19 days for Group 2, and 14 days for Group 1. The 25-day total water withhold period was chosen based on a previous droughting trial that showed significant drought induced mortality starting at this time period. There were three fewer Douglas-fir saplings in the control group due to pre-study mortality. Prior to drought treatments, average (±SE) root collar diameters were 1.7 ± 0.03 cm and 2.1 ± 0.05 cm, and mean heights were 0.82 ± 0.02 m and 1.0 ± 0.02 m for *P. monticola* and *P. menziessii*, respectively.

**Table 1.** Sample size for each drought stress group showing average foliar moisture content (SD) for each group.

| Species | Control—0 Days | | Group 1—14 Day Drought | | Group 2—19 Day Drought | | Group 3—25 Day Drought | |
|---|---|---|---|---|---|---|---|---|
| | FMC (%) | Sample Size | FMC (%) | Sample Size | FMC (%) | Sample Size | FMC (%) | Sample Size |
| western white pine | 173.69 (17.61) | 14 | 168.78 (19.97) | 16 | 158.25 (18.33) | 16 | 52.40 (55.09) | 16 |
| Douglas-fir | 148.90 (10.47) | 13 | 158.53 (19.32) | 16 | 103.23 (57.62) | 16 | 19.92 (15.70) | 16 |

### 2.2. Data Collection and Processing

For each of the 123 saplings, we collected foliar spectral reflectance using an ASD FieldSpec Pro spectroradiometer (Malvern Panalytical Ltd., Malvern, UK) equipped with the mineral probe attachment. Each of these measurements represents the average of 10 measurements that the instrument rapidly collects. This spectroradiometer collects measurements at wavelengths between 350 and 2500 nm and has a spectral resolution of 3 nm between 350 and 1000 nm and 10 nm between 1000 and 2500 nm. Prior to export, the data are resampled to 1 nm by the instruments' software, resulting in 2151 spectral bands. Three measurements were acquired in the top 1/3 of the canopy of each sapling, each being the result of 10 rapidly averaged collections by the ASD probe. For each spectral sample, ~5 cm$^2$ of foliage was positioned between a background object of known reflectance and a mineral probe attachment. Radiance measurements were calibrated using a 100% reflective Lambertian Spectralon panel (Labsphere Inc., North Sutton, NH, USA) prior to the measurement of each new sapling, following Sparks et al. [22]. During the processing of each sapling, ~5 g of needles were collected randomly throughout the top 1/3 of the canopy and immediately had their wet sample weight recorded (±0.01 g). These foliar samples were then oven-dried for 24 h at 100 degrees C and weighed again to acquire their dry weight. Finally, FMC was calculated using the difference between the dry and wet weights of the needles (Equation (1), Figure 1) [4].

$$\text{Foliar Moisture Content} = \frac{\text{wet weight (g)} - \text{dry weight (g)}}{\text{dry weight (g)}} \times 100 \qquad (1)$$

The three canopy reflectance measurements acquired for each sapling were averaged (Figure 2) and converted to band equivalent reflectance (BER) [33,34] of the 10 spectral bands of the MicaSense Dual-Camera system through convolving the spectra with the percent transmissivity values associated with these bands. This transmissivity function enabled the calculation of a single reflectance value for each band representative of the theoretical reflectance value that would be sensed by the MicaSense Dual-Camera system [27]. Using the 10-band values for each sapling, we calculated several spectral indices that have been demonstrated to be useful for predicting various measures of vegetation health (Table 2). Additionally, one novel index was tested for this study, the foliar moisture content index (FMCI), by examining the gaps in the spectral response curves of drought-stressed versus healthy saplings using similar methods to those employed by

Gao (1996; Table 2, Figure 3) [15]. Specifically, we utilized the spectral separation between healthy and drought-stressed vegetation present in the red edge 3 and NIR channels. This provided 20 predictor variables to test in our models (tree species, 10 spectral bands, and 9 spectral indices).

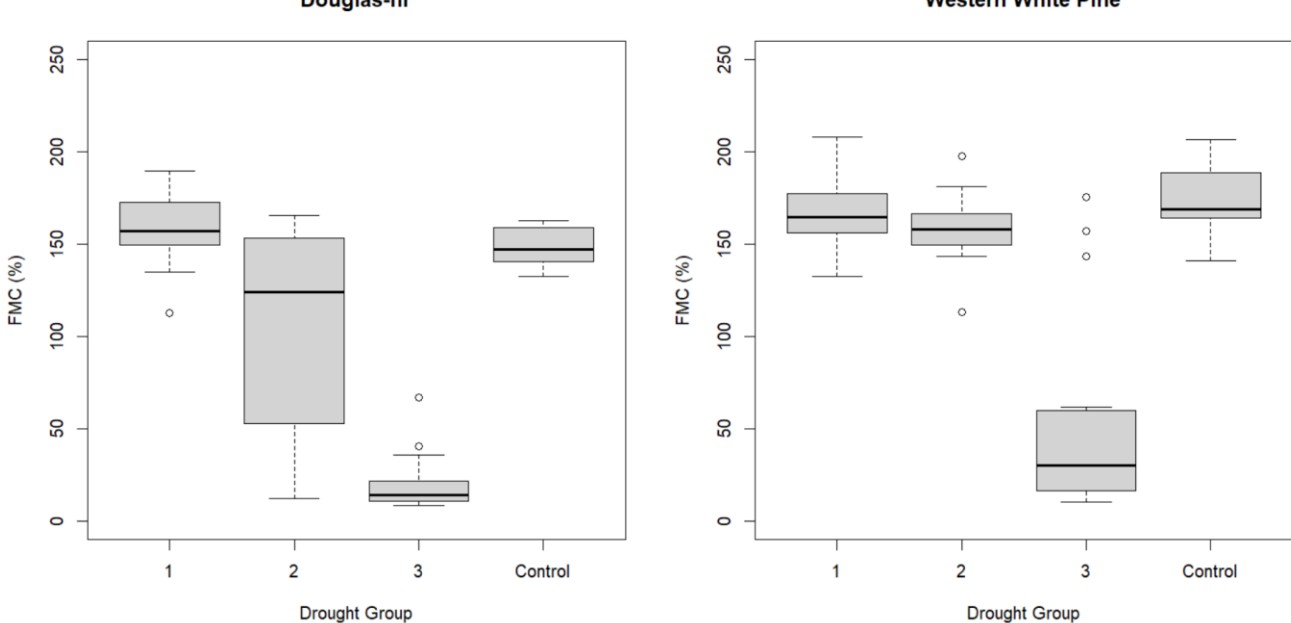

**Figure 1.** Boxplots of foliar moisture content (FMC; %) for each drought group and each species. Within each boxplot, horizontal lines represent the median, the box shows the first and third quartiles, the whiskers represent the maximum and minimum values within 1.5 times the interquartile range, and dots represent outliers.

**Table 2.** Spectral indices derived from the 10 spectral bands available with the MicaSense Dual Camera system, along with the formulation and examples of their previous application.

| Index | Index Name | Equation | Main Application |
|-------|-----------|----------|------------------|
| NDVI | Normalized Difference Vegetation Index | (NIR − Red)/(NIR + Red) | Chlorophyll content/Plant greenness [35] |
| NDVI2 | Normalized Difference Vegetation Index 2 | (NIR − Red2)/(NIR + Red2) | Not commonly used; Chlorophyll content/Plant greenness |
| GNDVI | Green Normalized Difference Vegetation Index | (NIR − Green)/(NIR + Green) | Photosynthetic activity/greenness [36] |
| GNDVI2 | Green Normalized Difference Vegetation Index 2 | (NIR − Green2)/(NIR + Green2) | Not commonly used |
| NDRE | Normalized Difference Red Edge | (NIR − Red Edge1)/(NIR + Red Edge1) | Plant health of mature plants [37] |
| GRVI | Green Ratio Vegetation Index | NIR/Green | Phenological indicator [38] |
| NDWI | Normalized Difference Water Index | (Green − NIR)/(Green + NIR) | Water content of water bodies [16] |
| PRI | Physiological Reflectance Index | (Green2 − Green)/(Green2 + Green) | Crop health monitoring [39] |
| FMCI | Foliar Moisture Content Index | (Red Edge3 − NIR)/(Red Edge3 + NIR) | Developed for this study |

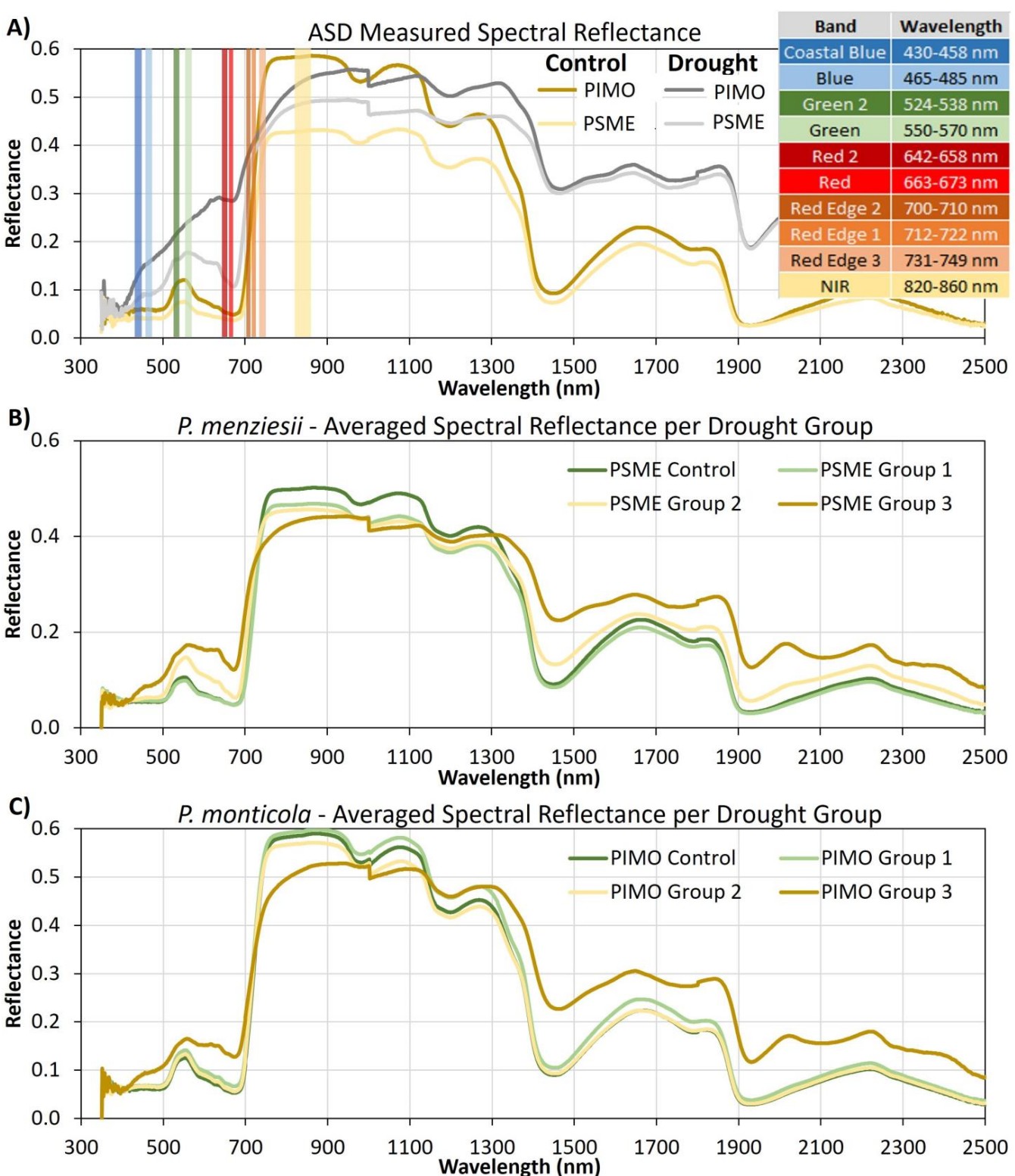

**Figure 2.** Spectroradiometer-measured foliar spectral reflectance of (**A**) two saplings in the droughted groups in gray (*P. monticola* FMC: 10.2%; *P. menziesii* FMC: 8.8%) and two saplings in the control group in yellow (*P. monticola* FMC: 140.5%; *P. menziesii* FMC: 142%). The vertical-colored bars show the 10 spectral bands available on the Micasense Dual-Camera sensor. The associated table colors match the spectral bands and report the range of wavelengths covered by each band. (**B**) Averaged spectral reflectance of each drought group for western white pine. (**C**) Averaged spectral reflectance of each drought group for Douglas-fir.

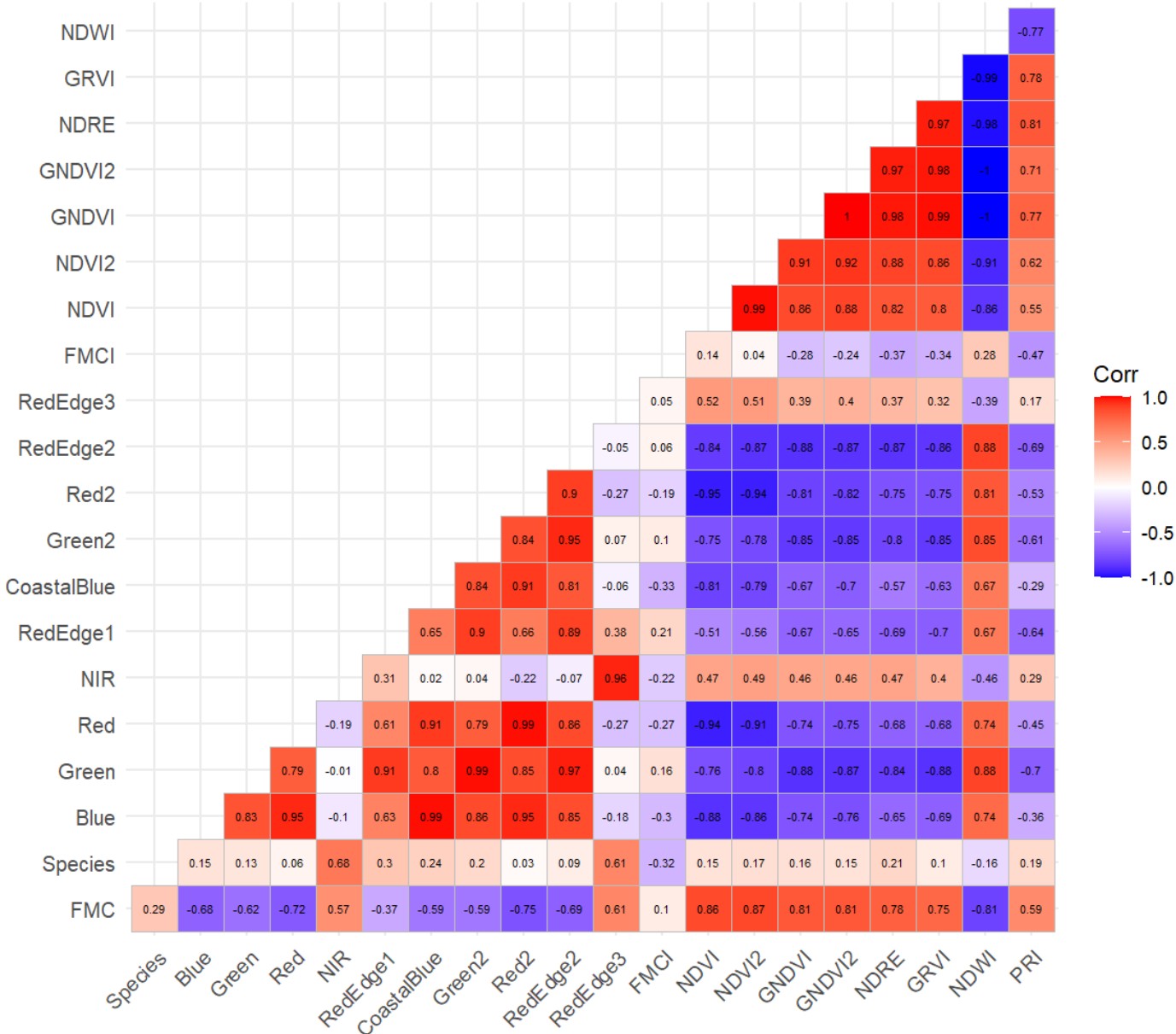

**Figure 3.** Correlation between the 20 predictor variables and foliar moisture content (FMC). For variables with >70% correlation we removed the variables with lower correlation to FMC.

### 2.3. Model Development

We developed classification and regression-based models to predict drought stress and FMC using the spectral predictor variables as inputs. Prior to model development, we examined predictor variable collinearity (Figure 3), and for predictor pairs that had a correlation of >70%, we removed the variable with lower correlation with FMC. The remaining predictors were evaluated for variable importance using the varImpPlot function in the randomForest package [40] within R statistical software version 4.1.0 [41,42]. This two-step process resulted in five predictor variables which included (in order of importance): NDVI2, red edge 3, PRI, FMCI, and species. For all models, both classification and regression, input data were randomly split into training (70%) and validation (30%) subsets.

Both a random forest classification model (RFCM) and a logistic classification model (LCM) were developed using the training dataset to predict whether saplings were drought-stressed or healthy based on a threshold of above or below 120% FMC. This FMC threshold was used as it represents the approximate moisture content where shrub and tree foliage become more receptive to fire spread [43,44]. We tested these models as the RFCM allows

for increased model complexity while the LCM provides more interpretable results. The RFCM was developed using the randomForest package [40] in the R statistical software. The RFCM was run using the five selected predictor variables, with seed 1002 and using 199 decision trees. The LCM was developed using base R and the glm function [41]. Variable selection for the LCM was performed using forward and backward stepwise selection to identify the subset of variables that minimized the Akaike Information Criterion (AIC) in the MASS package [45]. We assessed classification accuracy using confusion matrices calculated between the validation dataset and the model classification results. We report three commonly used accuracy metrics: overall accuracy, omission error and commission error.

Predictions of FMC as a continuous response were made by developing a set of simple linear models (SLRM), a multiple linear model (MLRM), and a random forest regression model (RFRM). For the SLRM, we compared five univariate models using each predictor variable: NDVI2, red edge 3, FMCI, PRI, and species. For the MLRM, variable selection was conducted using ordinary least squares forward and backstep stepwise selection to identify the variable subset that minimized AIC from the olsrr package in R [46]. Bootstrapping with 100 repetitions was used to resample the training and validation data to avoid overfitting for all of the SLRMs and the MLRM. The RFRM was developed following the same steps as the classification version, with the only difference being the use of FMC as a continuous response variable. Model performance was assessed using linear regression analysis between the observed FMC and model predicted FMC. Residual standard error and the coefficient of determination ($r^2$) were computed and used to evaluate the relationship 'goodness of fit'.

## 3. Results

### *3.1. Foliar Moisture and Spectral Variation*

Visual comparison of the FMC between the species and across the drought groups shows similar trends with a few small differences (Figure 1). The western white pine had both wetter and more variable FMC in all but Drought Group 2 compared to the Douglas-fir. A more continuous distribution of FMC was achieved by the Douglas-fir drought groups, largely due to the wide range of values in Drought Group 2 (Figure 1). However, the western white pine resulted in a wider total range of FMC, but with a notable gap in FMC values from ~60% to ~125%.

The spectral response curves varied between species, with western white pine having higher reflectance from ~750–1300 nm compared to Douglas-fir for all drought groups (Figure 2B,C). However, the mean reflectance of drought groups varied more for Douglas-fir than western white pine. Spectral separation was only apparent in the green (550–570 nm) and green 2 (524–538 nm) bands for the longest duration drought group (Group 3). For both species, the greatest spectral separation across the drought groups occurred in the NIR (820–860 nm), followed by the red (663–673 nm) and red 2 (642–658 nm) bands (Figure 2). Although outside the spectral bands tested in this study, the shortwave infrared portion of the spectrum also exhibited strong separation of the drought groups for both species.

Evaluation of the 20 predictor variables for collinearity showed that NDVI2 was the strongest single predictor, but that it was highly colinear with several other terms (Figure 3). Because of high correlation (>|70%|) with NDVI2, the following predictors were removed from analysis, including coastal blue, blue, green, green 2, red, red edge 2, NDVI, GNDVI, GNDVI2, NDRE, GRVI, and NDWI.

### *3.2. Classification Models*

The RFCM retained the NDVI2, PRI, red edge 3, and FMCI. In the final set of predictor variables, species was removed due to its low variable importance. The RFCM had an $R^2$ of 85.41 and produced a mean square error (MSE) of 0.049, with balanced classification errors between the two classes (Table 3). When the RFCM was tested against the validation data, the model resulted in a final overall classification accuracy of 94.44%, with drought-stressed

saplings having an omission error of 9.09% and a commission error of 9.09%, while healthy saplings had an omission error of 4.0% and a commission error of 4.0%.

**Table 3.** Confusion matrices of the predicted versus observed class for the random forest classification model (RFCM) and logistic classification model (LCM) using the validation data. Overall accuracies were 94.44% and 97.22% for the RFCM and LCM, respectively.

| Random Forest | | Reference Class | | |
|---|---|---|---|---|
| | | **Drought-Stressed** | **Healthy** | **Commission Error** |
| Predicted Class | Drought-Stressed | 10 | 1 | 9.09% |
| | Healthy | 1 | 24 | 4.00% |
| | Omission Error | 9.09% | 4.00% | |
| Logistic Regression | | | | |
| Predicted Class | Drought-Stressed | 11 | 0 | 0.00% |
| | Healthy | 1 | 24 | 4.00% |
| | Omission Error | 8.33% | 0.00% | |

Prediction accuracy of the LCM was 96.55% for the training data (Table 3) and, when tested against the validation data, an overall classification accuracy of 97.22% was achieved. Omission errors of 8.33% for the drought-stressed saplings and 0% for the healthy saplings and commission errors of 0% for the drought-stressed saplings and 4.0% for the healthy saplings were observed. When developing the LCM, species was the only predictor variable removed during the stepwise variable selection process (Table 4). Since a true $r^2$ cannot be calculated for LCMs, we calculated McFadden's $r^2$, which has values ranging from 0 to 1, and achieved a value of 0.84. Any McFadden $r^2$ values over 0.4 demonstrate a strong model fit to the data and high predictive power [47]. The RFCM and the LCM misclassified the same sapling, with the RFCM misclassifying an additional sapling (Table 3). One drought-stressed Douglas-fir sapling (FMC: 112%) was confused as healthy by both models. For the RFCM, one healthy western white pine sapling (FMC: 132%) was confused as drought-stressed.

**Table 4.** Logistic classification model (LCM) coefficients. The Z-value is the regression coefficient divided by the standard error.

| | **Coefficient** | **Standard Error** | **Z-Value** | ***p*-Value** |
|---|---|---|---|---|
| Intercept | 43.63 | 18.07 | 2.41 | <0.05 |
| NDVI2 | −11.68 | 7.50 | −1.55 | 0.12 |
| Red edge 3 | −89.16 | 39.14 | −2.28 | <0.05 |
| FMCI | −115.01 | 53.35 | −2.16 | <0.05 |
| PRI | −147.73 | 56.94 | −2.60 | <0.05 |

### 3.3. Regression Models

The best-performing SLRM used NDVI2 as the predictor variable and resulted in an adjusted $R^2$ of 73.41 ($p < 0.05$) with a residual standard error of 33.08 (Table 5). For this model, each 0.1 increase in NDVI2 resulted in a 26.0% increase in FMC. Comparison of predicted SLRM versus observed FMC showed strong general relationships but with the greatest prediction errors in the middle of the range of FMC values (Figure 4A). The NDVI2 SLRM explained more than twice as much of the variation in FMC compared to any of the other SLRMs tested.

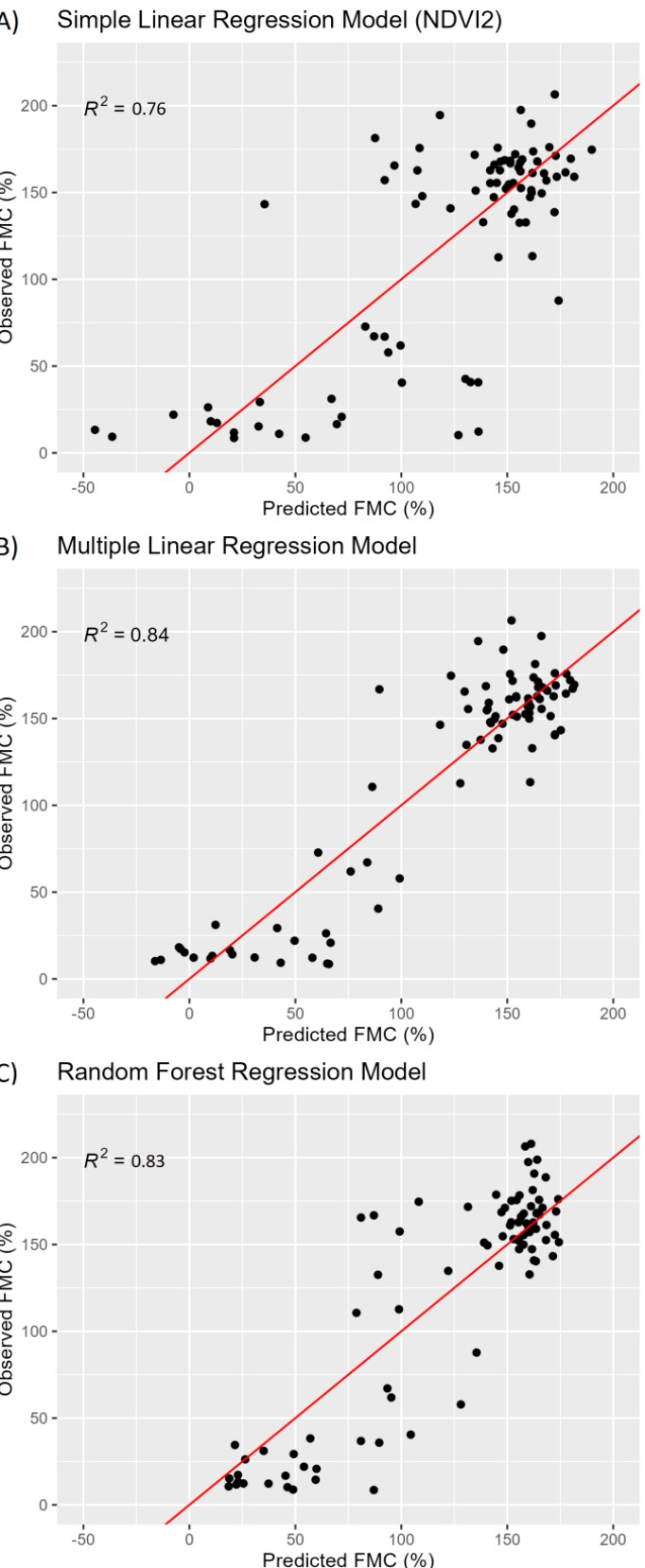

**Figure 4.** Predicted versus observed foliar moisture content (FMC) for (**A**) the simple linear model (SLM) using NDVI2, (**B**) the multiple linear regression model (MLRM), and (**C**) the random forest regression model. The red line in each panel shows the 1:1 relationship.

**Table 5.** Predictor variable results from five simple linear regression models (SLRM).

| Predictor Variable | Intercept | Coefficient (SE) | *p*-Value | Residual Standard Error | Adjusted r² |
|---|---|---|---|---|---|
| NDVI2 | 13.17 | 267.66 (17.47) | <0.05 | 33.08 | 73.41 |
| Red edge 3 | −135.46 | 1504.39 (223.94) | <0.05 | 52.32 | 33.91 |
| FMCI | 28.13 | 375.90 (323.54) | 0.25 | 63.18 | 0.41 |
| PRI | 158.16 | 1732.76 (255.01) | <0.05 | 52.12 | 34.44 |
| Species | 101.56 | 36.28 (13.19) | <0.05 | 61.52 | 7.09 |

The MLRM used 100 bootstrap repetitions for resampling, and the OLS stepwise variable selection informed the removal of species. The final MLRM removed species as a predictor and resulted in a final adjusted $r^2$ of 82.62 ($p < 0.05$) with a residual standard error of 26.77 (Table 6). When testing the model against the validation dataset, we achieved an $r^2$ of 84.57 ($p < 0.05$) with a residual standard error of 25.05. The MLRM represented an ~19% improvement in the model residual standard error over the best SLRM. Each of the spectral indices showed a positive response, indicating that increases in any of the spectral index values resulted in increased FMC prediction (Table 6). Comparison of MLRM predictions to observed values showed mostly normally distributed residuals throughout the data range, but 3.4% of samples were predicted to have negative FMC values (Figure 4B). The RFRM used NDVI2, red edge 3, FMCI, and PRI as predictors and had an $r^2$ of 75.32 and a root mean square error of 31.37 (Figure 4C). Inspection of the RFRM predicted versus observed values showed slightly greater variation in prediction accuracy in the middle of the dataset compared to the MLRM but eliminated the negative predictions seen from the MLRM (Figure 4C).

**Table 6.** Multiple linear regression model (MLRM) output table.

| | Coefficient | Standard Error | T Value | *p*-Value | AIC |
|---|---|---|---|---|---|
| Intercept | −226.61 | 41.18 | −5.50 | <0.05 | - |
| NDVI2 | 138.64 | 21.39 | 6.48 | <0.05 | 860.96 |
| Red edge 3 | 611.92 | 149.30 | 4.10 | <0.05 | 850.84 |
| FMCI | 851.31 | 159.19 | 5.38 | <0.05 | 833.78 |
| PRI | 1039.63 | 202.10 | 5.14 | <0.05 | 854.59 |

## 4. Discussion

The strongest-performing drought stress classification and FMC regression models were the LCM and the MLRM, with an overall accuracy of 96.55% and $r^2$ of 82.62, respectively. Both models used NDVI2, PRI, red edge 3, and FMCI as their predictor variables. The MLRM model explained 82.6% of the variation but produced a residual standard error of 26.7% FMC. While this error level is fairly consistent in an absolute sense throughout the tested range of FMC values, the relative magnitude of the error is going to be exacerbated at lower FMC levels. While the model captures the general trend in FMC, it may struggle to differentiate trees near the critical 120% FMC level. The high accuracy of the LCM indicates that a consumer-grade UAV sensor, such as the MicaSense Dual-Camera, could accurately predict the drought status of Douglas-fir and western white pine. The results also suggest that this approach could be used to assess the drought stress status of other tree species and potentially be scaled across landscapes in combination with coarser-resolution imagery

with further study. The MLRM could have applications where there is a need to assess conifer physiology, ignition risk, and/or susceptibility to insects and disease. The fact that the tested species represent both short- and long-needle conifers, which did not impact model performance, is promising for the potential transference of these relationships to other conifer species. However, more studies are needed with additional conifer species in both controlled laboratory and natural forest settings to understand if these relationships are similar across species and scale to older and larger trees.

The models developed in this study use existing spectral indices, and combined with the high model accuracy, suggests that UAVs equipped with similar sensors could improve the spatial resolution and scale of FMC predictions. To the authors' knowledge, no prior studies exist which quantify the foliar moisture content and drought stress of conifers using a UAS; however, Blanco et al. [48] were able to quantify cherry tree leaf water potential ($r^2 = 0.67$) using UAS-derived NDVI. Studies in agricultural systems have achieved root mean square error values as low as 1% when predicting maize FMC using data from a similar UAS multispectral sensor to that tested in this study [49]. Previous studies predicting FMC in grasslands have achieved $r^2$ values of 0.91 using MODIS imagery at 500 m spatial resolution [50], while models of FMC using hyperspectral data have predicted FMC of forest canopies using the normalized difference infrared index and achieved an $r^2$ of 0.9 [51]. However, these hyperspectral data, part of the NASA HyspIRI Mission, have not yet been launched, precluding their immediate application [51]. While a range of success has been found in predicting FMC across various ecosystems from moderate resolution sensors [52–56], these sensors are unable to provide the individual tree-level information needed for management decisions related to promoting resistance and resilience to disturbances at the stand-level.

Our MLRM and LCM models both retained NDVI2, PRI, red edge 3, and FMCI as significant predictors. NDVI2 and FMCI are not widely used indices and the authors could not identify any papers using either index for tree health assessments. However, NDVI2 is only slightly different from NDVI in that it uses the second red band (red 2) available on the MicaSense Dual-Camera sensor, which is narrower and covers the lower part of the Sentinel-2a red band. This narrowing of the band spectral range might account for the increased performance over satellite-based observations as changes in reflectance are averaged over smaller channels. Similar to NDVI, we would expect NDVI2 to have higher values for vegetation with lower red reflectance, assuming NIR reflectance remains constant. NDVI2 uses lower wavelengths of red light and occurs in a region where there is a larger difference in reflectance between healthy and drought-stressed saplings (Figure 2). This narrower band allows us to detect smaller changes in moisture, similar to the increased accuracy seen by Hunt et al. [51] when using hyperspectral imagery. Since FMCI was developed for this study by examining the spectral reflectance signatures of a drought-stressed and healthy sapling, it was expected that FMCI would help predict continuous and classified FMC response variables. However, FMCI had lower explanatory power for predicting FMC than the other final four predictors. This reduced explanatory power is likely due to inconsistent shifts in the red edge 3 band between the two species across the range of FMC. PRI was originally developed for the assessment of agricultural vegetation vigor and moisture and appears to have significant explanatory potential for conifer tree health status [24,57]. Sensors that incorporate short-wave infrared bands would likely result in improved accuracy in predicting FMC as shortwave infrared reflectance is highly responsive to water content in foliage [15,58]. However, current cameras for UAS that operate in the SWIR are costly, potentially limiting the ability for managers to acquire such sensors. As these sensors become more broadly available it would be worth considering SWIR bands in the prediction of FMC from UAS platforms. Such bands when applied through hyperspectral remote sensing have provided substantial improvements in predicting FMC [51]. Finally, the red edge 3 band covers a spectral range not available on conventional satellite-based sensors. This spectral range has high reflectance for healthy saplings and lower reflectance for drought-stressed saplings but, compared to other bands and indices used in this study,

has relatively low variance across our samples (range of 0.13). Since light is reflected by living vegetation in this red edge 3 channel, consistent differences in reflectance represent potential biologic indicators that can be exploited for predicting FMC.

Modeling results suggest that UAS equipped with sensors like the MicaSense Dual-Camera system could be used for identifying vegetation stress as it relates to changes in FMC. In prescribed fire planning and fire behavior modeling, FMC is an important input and influences manager expectations of fire effects [59]. Additionally, drought-status has a strong correlation in many conifer species with bark beetle susceptibility [60]. Since FMC varies across and within vegetation types and within individual tree crowns, the flexibility of our random forest models may be beneficial for capturing this range of conditions. While our random forest models performed slightly worse than the logistic and multiple linear models, most other UAS studies of foliar moisture have shown that machine-learning strategies achieve the highest model accuracy [49]. Operational field deployment of these methods will need to be flexible enough to overcome vertical and horizontal gradients in FMC within individual tree crowns associated with shadowing and solar angles that impact both reflectance and FMC. Our models were developed based on an average of three FMCs collected randomly across sapling tree crowns. In mature forests, data summarization strategies will need to be explored to account for variation in the multiple pixels within a single tree crown to accurately characterize the mean or median FMC. Within western US dry conifer forests, reasonable success has been achieved at delineating individual tree crowns from UAS-derived canopy height models [61]. The resulting individual tree crown polygons could be used to isolate the spectra for a single tree for applying models similar to those developed in this study. Successful deployment of such individual tree drought monitoring models holds potential to inform forest thinning operations aiming to improve forest drought resilience. Future work should focus on applying a suite of predictive models in mature forests across a range of site conditions to examine the ability of different models to reliably predict individual tree crown FMC.

## 5. Conclusions

Using spectroradiometer-derived band equivalent reflectance of a multispectral consumer-grade UAS camera, we were able to accurately predict FMC and drought stress status (FMC < 120%) for western white pine and Douglas-fir saplings. While these models were developed using spectroradiometer derived data, the high FMC prediction accuracy using band equivalent reflectance of a UAS sensor suggests that accurate FMC quantification using UAS is possible; however, this study should be repeated using UAS-collected imagery. The logistic model developed in this study to classify stressed and non-stressed saplings had high overall classification accuracy for both species and could be beneficial for land managers seeking to incorporate drought status information when planning prescribed fire or other silvicultural treatments. Similarly, the multiple linear regression model used to predict FMC could aid efforts needing rapid and spatially extensive FMC observations of saplings or mature trees to understand patterns of drought stress within and among forest stands. Specifically, this model could aid in assessing tree-level physiological responses to different growing environments or treatment designs. While the FMC results in this study are promising, these models need further testing using UAS-derived data across a range of tree sizes and species to determine their applicability and transferability across conifer forests experiencing drought.

**Author Contributions:** This project was designed by L.E.L., W.T.T. and A.M.S.S. L.E.L. processed, analyzed, and interpreted the data and prepared the manuscript with the help of W.T.T. Data collection and the laboratory setup was led by A.M.S.S. and A.M.S., who provided feedback and edits on the manuscript. All authors have read and agreed to the published version of the manuscript.

**Funding:** Lad is funded under the USDA Hatch (COL00401) and JFSP GRIN (23-1-01-24) programs. Smith is partially funded by the National Science Foundation under award #2242769 and Smith and Sparks by the USDA NIFA program under award 2022-10955. Partial funding for Sparks was also provided by the National Institute of Food and Agriculture, USDA, McIntire Stennis project under IDAZ-ES-0609. This publication was supported by the USDA Forest Service, Rocky Mountain Research Station. The findings and conclusions in this publication are those of the authors and should not be construed to represent any official USDA or U.S. Government determination or policy.

**Data Availability Statement:** The data presented in this study are available on request from the corresponding author.

**Conflicts of Interest:** The authors declare no conflict of interest.

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
