# Peer review of "Evaluating Predictive Models of Tree Foliar Moisture Content for Application to Multispectral UAS Data: A Laboratory Study"

_remotesensing, doi:10.3390/rs15245703_

Round 1
Reviewer 1 Report
Comments and Suggestions for Authors
The article titled " Evaluating Predictive Models of Tree Foliar Moisture Content for Application to Multispectral UAS Data: A Laboratory Study" presents an important approach to accurate sapling drought-stress detection using uncrewed aerial system (UAS) images, with far-reaching implications for management decisions related to promoting resistance and resilience to disturbances. The authors identify a gap in existing methods' accuracy when detecting the foliar moisture content by satellite remote sensing, which is due to low spatial and temporal resolution. They demonstrate the utility of fine-scale laboratory experiments for examining linkages between spectral indices derived from foliar spectral reflectance and plant physiological metrics. So, they propose a novel system " UAS Data", that combines fine-scale laboratory, field sampling, and satellite-derived landscape-scale assessments.
The importance of the proposed method lies in the multiple canopy reflectance measurements (10 measurements for each sapling for 123 total saplings ) at wavelengths between 350 and 2500 nm. The important point was to convert these reflectance measurements to band equivalent reflectance (BER) using the 10-band values for each sapling. This study tested a fair number of predictors (several spectral indices and spectral bands). Additionally, one novel index was created, the foliar moisture content index (FMCI), by examining the gaps in the spectral response curves of drought-stressed. A random forest classification model (RFCM) and a logistic classification model (LCM) were developed to predict whether saplings were drought-stressed or healthy based on a threshold of above or below 120% FMC.
A major highlight is the rigorous validation process involving the UAS data. The authors assessed classification accuracy using confusion matrices: overall accuracy, omission error and commission error. Model performance was assessed using linear regression analysis between the observed FMC and model predicted FMC. Residual standard error and the coefficient of determination (r2) were computed and used to evaluate the relationship. A performing drought stress classification and FMC regression model were obtained with a final overall classification accuracy of 94.44% for RFCM and 97.22 % for LCM, while the adjusted r2 of 82.62 for MLRM which used NDVI2 as a variable
Notably, FMC model's applicability could extend to a range of challenging scenarios as exemple : assessing tree-level physiological responses to different growing environments, understanding patterns of drought stress within and among forest stands, providing information for planning prescribed fire or other silvicultural treatments. The model consistently demonstrates an important role across these contexts, highlighting its applicability at large scale.
In conclusion, this article contributes significantly to the field of tree foliar moisture content prediction based on laboratory measurements. However, it is not sure whether it is possible to obtain the same promising results when applying these models to conifer forests of different ages and sizes in using UAS-collected imagery.
(1) Summary of the Paper's Aim, Contributions, and Strengths:
The paper introduces a novel model to predict FMC and sapling drought-stress. The overall objective is to test if the band equivalent refelcetnce (BER) of a consumer-grade multispectral UAS camera could be used to to predict FMC for western white pine and Douglas-fir, a crucial task for fire management and ecology assessment. The main contributions lie in the innovative integration of the laboratory experiments that provide accurate information about varying levels of drought stress with field measurements and UAS data, which could provide rapid, timely information for management decisions. The paper demonstrates the possibility of obtaining performance models to predict the FMC using RFCM and MLRM based on NDVI2, which forme the basis for expanding research to include different species and forests of different sizes and ages.
(2) General Concept Comments:
(2) General Concept Comments:
The paper presents a compelling approach to enhancing FMC prediction in UAS images. However, several areas warrant clarification and consideration. Firstly, the study is based on 3 drought stress groups but the research does not clarify whether the resulting models were sensitive to the drought stress group. It is also not explained why the stress period ends at 25 days, can't obtain longer drought periods? that are more representative to the drought periods at forest scale. Additionally, the proposed index (FMCI) didn’t give the expected results and did not discuss why ? the authors did not also clarify the spatial resolution of their data, nor the formula of the model obtained.
(3) Specific Comments:
- Line 144: " A total of 63 western white pine and 60 Douglas-fir saplings" -it is not related with the table 1.
- Line 190: "This provided 21 predictor variables to test in our models " - it is not related with the Figure 3, which shows only 19 predictor
- Table 3: overall accuracy, omission error and commission error do not appear in the confusion matrix
- Figure 3: Correlation between the 21 predictor variables (i see just 19 ?) and foliar moisture content (FMC). The table shows that the correlation between FMC and red edge 3, PRI, FMCI, and species <70%, while the correlation between FMC and bleu, red, red2, NDVI NDVI2, GNDVI, GNDVI2, NDRE, GRVI and NDWI >70%. Does the figure show a typographical error or confusion about the matter especially, that the model the more robust is based on NDVI2 between the 5 selected predictor (FMC and red edge 3, PRI, FMCI and NDVI2). Whereas the rest of the predictors did not show a strong correlation with FMC
These comments aim to enhance the clarity, rigor, and depth of the paper's scientific content, facilitating a more comprehensive understanding of FMC performance models and potential areas for future research.
Author Response
Thank you for taking the time to review our paper. Please see the attachment.

Reviewer 2 Report
Comments and Suggestions for Authors
The study of the trees foliar moisture content is novel, as mentioned in the manuscript. However, there are some recommendations, which are listed below:
1. Include in the section Introduction a literature review on the state of the art of UAS application and classification and regression algorithms or other techniques in the determination of foliar moisture content in trees or other agricultural crops.
2. It should be briefly explained how the FMCI index was obtained or place the reference bibliography.
3. Lines 314-329 could be included in the Introduction section.
4. Large standard error values are shown in the Results, however, they are not discussed in the corresponding section, their discussion should be considered.
Author Response

(The authors gave the same response as above.)

Reviewer 3 Report
Comments and Suggestions for Authors
The article is an experimental study aimed at identifying the stress level of trees based on the water content of needles, estimated from visible to SWIR reflectance spectra.
The study was carried out on young trees subjected to artificial water stress. The data is analysed and statistical models are proposed for interpreting the measurements, making it possible to classify a dry state and a wet state or to estimate the FMC quantitatively.
The data are rich and the statistical approaches well and relevant. However, the statistical approach raises the question of how to models (to older trees with different architectures, to other species). This point could be discussed more deeply.
One of the original features of this work is that it deals with the water content of leaves in a forest environment. Although there are few studies of this, the authors refer to the study by Hunt et al [48]. It would have been interesting and applied the same approach as a benchmark.
Reflectance in the SWIR is often put forward as an interesting band for water content. t is surprising that no indicator using bands in this area have been used. This could have been addressed at least in the discussion.
The authors give some illustrations ofreflectance spectrum. These illustrations are really essential and it would have been interesting to present more results, focusing on the evolution of the spectraover a whole range of FMC or the variability of the measurements for given tree.
Having said that, these comments do not deny the quality of the work, which deserves to be published. In addition to the above points, which might enrich the text and its scope, I have a few minor comments:
L43 I'm not sure that the link between leaf moisture and bark beetle and susceptibility to the bark beetle is very direct, the tree's defences being particularly being particularly linked to the tree's ability to produce resin to defend itself.
L78; Sentinel's revisit time is more like 5 days
L163: Is it possible to have a spatial resolution of the measurements
L163: The acquisition protocol is not clear: status of the 10 measurements vs the 3 measurements acquired on the top of the canopy (L167)
Table 4: Z value needs to be defined
L329 tree-level information: unclear
Author Response

(The authors gave the same response as above.)
